# Three-Stage Framework for Accurate Pediatric Chest X-ray Diagnosis Using Self-Supervision and Transfer Learning on Small Datasets

**DOI:** 10.3390/diagnostics14151634

**Published:** 2024-07-29

**Authors:** Yufeng Zhang, Joseph Kohne, Emily Wittrup, Kayvan Najarian

**Affiliations:** 1Department of Computational Medicine and Bioinformatics, University of Michigan, Ann Arbor, MI 48109, USAkayvan@med.umich.edu (K.N.); 2Department of Pediatrics, University of Michigan, Ann Arbor, MI 48103, USA; 3Michigan Institute for Data Science (MIDAS), University of Michigan, Ann Arbor, MI 48109, USA; 4Department of Emergency Medicine, University of Michigan, Ann Arbor, MI 48109, USA; 5Max Harry Weil Institute for Critical Care Research and Innovation, University of Michigan, Ann Arbor, MI 48109, USA

**Keywords:** chest X-ray, medical image analysis, self-supervised learning, transfer learning, model interpretability

## Abstract

Pediatric respiratory disease diagnosis and subsequent treatment require accurate and interpretable analysis. A chest X-ray is the most cost-effective and rapid method for identifying and monitoring various thoracic diseases in children. Recent developments in self-supervised and transfer learning have shown their potential in medical imaging, including chest X-ray areas. In this article, we propose a three-stage framework with knowledge transfer from adult chest X-rays to aid the diagnosis and interpretation of pediatric thorax diseases. We conducted comprehensive experiments with different pre-training and fine-tuning strategies to develop transformer or convolutional neural network models and then evaluate them qualitatively and quantitatively. The ViT-Base/16 model, fine-tuned with the CheXpert dataset, a large chest X-ray dataset, emerged as the most effective, achieving a mean AUC of 0.761 (95% CI: 0.759–0.763) across six disease categories and demonstrating a high sensitivity (average 0.639) and specificity (average 0.683), which are indicative of its strong discriminative ability. The baseline models, ViT-Small/16 and ViT-Base/16, when directly trained on the Pediatric CXR dataset, only achieved mean AUC scores of 0.646 (95% CI: 0.641–0.651) and 0.654 (95% CI: 0.648–0.660), respectively. Qualitatively, our model excels in localizing diseased regions, outperforming models pre-trained on ImageNet and other fine-tuning approaches, thus providing superior explanations. The source code is available online and the data can be obtained from PhysioNet.

## 1. Introduction

Pediatric respiratory diseases, such as pneumonia, are the leading cause of hospitalizations and mortality in young children or infants in the United States and across the whole world [1] with an estimated annual mortality of more than 0.7 million children under the age of five in the United States [2]. Early diagnosis and treatment can greatly reduce the risk of severe outcomes and mortality. Chest X-rays (CXRs) are the most commonly performed imaging examination in pediatrics and can aid in performing non-invasive, accurate interpretations and diagnoses and timely interventions in pediatric respiratory disease cases. Therefore, early thorax disease detection using a CXR is essential in pediatric medical research and practice.

Recent advances in deep learning (DL) have revolutionized the analysis of CXRs, with convolutional neural networks (CNNs) showcasing exceptional capabilities in diagnosing diseases and identifying affected regions [3,4,5]. Building on this success, newer architectures such as transformer-based and graph neural network-based models are gaining traction [6,7,8,9]. These models not only maintain a high diagnostic performance but also enhance the interpretability of the results, which is crucial for clinical acceptance. These advancements promise to sustain momentum towards more sophisticated, automated medical imaging diagnostics which will transform patient outcomes through technology.

Despite the advantages of CXRs and the significant advancement in applying deep learning to building decision support systems using CXRs, the development of accurate and efficient models hinges on the availability of precise annotations. However, accurately annotating CXRs remains a challenging task, even for skilled radiologists, due to the high costs and extensive time required for obtaining annotations. Furthermore, inconsistencies are common among physicians. To address these challenges, researchers are turning to self-supervised learning strategies such as BYOL [10], DINO [11], and Masked Autoencoders (MAE) [12]. These techniques enable the pre-training of models on large unlabeled datasets, followed by fine-tuning using smaller, task-specific labeled datasets. As demonstrated in recent studies [13,14,15], not only are these methods resource-efficient in terms of CXR label usage, but they also excel in developing robust image representations that often match or even surpass the performance of fully supervised approaches. A summary of self-supervised methods for CXR pre-training is presented in Table 1.

However, a notable limitation of self-supervised learning is its dependency on large datasets. The majority of existing large-scale CXR datasets primarily comprise adult cases [21,22], and there is a moderate difference in clinical characteristics between adult and pediatric patients [23]. Pediatric CXR datasets, particularly publicly available ones, are scarce and typically only contain samples for one specific disease [24]. The most comprehensive public pediatric CXR dataset to date has recently been released, containing only a few thousand images [25]. This data scarcity poses a challenge in developing accurate and reliable models for pediatrics despite ongoing research efforts.

In this study, we aim to create an accurate, automated system for diagnosing pediatric respiratory diseases from CXR, addressing the challenge of limited pediatric CXR data. To achieve this, we developed a three-stage transfer learning approach specifically for classifying pediatric CXR diagnoses, as shown in Figure 1. This approach involves (1) initially pre-training the system using a masked autoencoder, a self-supervised learning framework, on large-scale datasets of publicly available adult CXRs or natural images, (2) subsequently fine-tuning the model with additional adult CXR data, and (3) further fine-tuning with pediatric CXR data to enhance model accuracy and effectiveness in this specialized domain. We investigate the impact of pre-training the model on natural images or adult CXRs and the effects of fine-tuning it on various CXR datasets. This work is of high importance because there is little work focusing on pediatric CXRs due to the data availability and heterogeneity between adults’ and children’s CXR. In summary, the main contributions of our work include the following:We developed and evaluated a three-stage training framework specifically for the pediatric chest X-ray dataset.Our study compares this three-stage training framework to other approaches, demonstrating that the ViT-Base/16 model, pre-trained on CXR, fine-tuned on CheXpert, and further fine-tuned on PediCXR, outperforms the otheres.We examined the top-performing model’s ability to detect common diseases accurately.


## 2. Materials and Methods

### 2.1. Dataset

#### 2.1.1. Adult CXR Dataset

Four public adult CXR datasets were utilized in the three-stage approach for (a) pre-training and (b) fine-tuning. The MIMIC-CXR dataset provides 243,334 image–text paired chest X-rays across fourteen classes [21]. The CheXpert dataset is a large collection of 191,028 frontal-view X-rays, focusing on five common lung diseases [22]. The COVIDx dataset includes over 300,000 images, categorized into four disease classes [26]. Lastly, the ChestX-ray14 dataset comprises 112,120 frontal view X-rays, spanning fourteen disease classes [27].

#### 2.1.2. Pedatric CXR Dataset

In this study, we utilized the recently released and currently largest pediatric CXR dataset, PediCXR [25,28] for pediatric chest thorax disease diagnosis. This dataset comprises a training set of 7728 images and a test set of 1397. Each X-ray image is manually annotated by experienced and board-certified radiologists. The entire PediCXR dataset encompasses 15 diagnoses, but the test set includes only 11. Following the strategy of [29], we grouped less common diagnoses into a category labeled ‘other disease’, resulting in six disease categories: broncho-pneumonia, bronchiolitis, bronchitis, pneumonia, no findings, and other diseases. Detailed information about the dataset is shown in Table 2. The images, initially in DICOM format, were converted into JPG format with the Python package Pydicom version 2.4 for further processing. Each original image, with dimensions of 1692 by 1255 pixels, was resized to 224 by 224 pixels. Following resizing, standardization of the images was implemented using mean values of (0.5056, 0.5056, 0.5056) and standard deviations of (0.252, 0.252, 0.252) across the three channels [30].
Figure 1**The overall workflow of PediCXR classification task**. It consists of three stages. (**a**) Pre-training stage: self-supervised learning is performed using MAE on natural images or adult CXRs. (**b**) Adult CXR fine-tuning stage: the trained encoder undergoes supervised learning with the adult CXR dataset. (**c**) Knowledge-transferring stage: the trained encoder is further linear-probed/fine-tuned on the PediCXR dataset for specific knowledge acquisition.
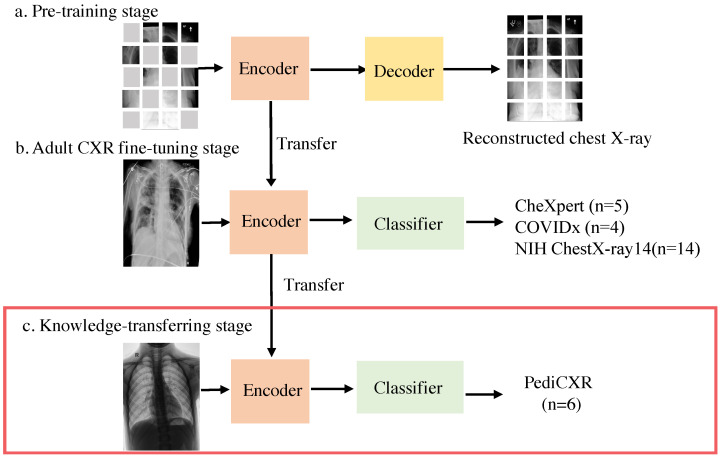



### 2.2. Vision Transformer, Masked Autoencoder and Transfer Learning

The transformer, based on the multi-headed self-attention mechanism and positional encoding, was first introduced in [31] for natural language processing and has shown its great power in multiple tasks. ViT [32] extends its application to the computer vision area. The images are split into patches and then fed into the transformer encoder to produce a representation for downstream classification. ViT outperformed ResNets by a significant margin on some tasks [33,34], showing its potential in CXR-related tasks.

The Masked Autoencoder (MAE) was first introduced in [12]. This approach incorporates both an encoder and a decoder. Random patches are masked based on a uniform distribution, and the remaining unmasked patches are processed and embedded using a standard ViT. Subsequently, the encoded patches and masked tokens are input into the decoder, which is tasked with predicting the masked information. Notably, the decoder’s role is confined to pre-training, enabling the encoder to learn image representations. The primary goal of MAE is to reconstruct the original image from the visible portions by predicting pixel values for the masked ones with a perceptual loss, similar to BERT, to learn the image’s representation. The effectiveness of MAE has been showcased across various image datasets [12,35] as well as in medical imaging datasets [30,36,37].

The encoder, pre-trained with MAE, can serve as a starting point for a related task via transfer learning. Transfer learning is a technique that transfers knowledge across different but related tasks, improving model performance and potentially reducing training time or training dataset size. In medical research, where data can be scarce, models are frequently pre-trained on one task and then fine-tuned for another related task. Using self-supervised pre-trained models as base encoders, transferring the learned weights from these self-supervised models to enhance performance is now popular and has been proven to be effective [38,39,40].

### 2.3. Training Strategy and Details

#### 2.3.1. Pre-Training Stage and Fine-Tuning Stage on Adult CXR

The MAE framework comprises an encoder and a simpler decoder. Two variants of the Vision Transformer (ViT) serve as encoders: ViT-Small/16 and ViT-Base/16. These encoders break down images into patches of non-overlapping 16 × 16 pixels and then embed them into a lower-dimensional space. The ViT-Small/16 has a smaller MLP size (dim = 384) and fewer heads in its attention layers (*n* = 6) in contrast to the ViT-Base/16 (dim = 768, *n* = 12). Both MAE variants utilize the ImageNet and a combined CXR dataset for pre-training, respectively. The combined CXR dataset includes CheXpert (191,028 images), MIMIC-CXR (243,334 images), and NIH ChestX-ray14 (75,312 images). Pre-training aims to learn the general image/CXR patterns. To further improve the model performance on the downstream tasks and learn the more detailed distribution of diseases with label guidance, each CXR pre-trained model was specifically fine-tuned on these three datasets. This approach follows the methodology detailed in [30], using their reported pre-trained and fine-tuned model weights. These two foundational training stages equip the encoder with skills for interpreting X-ray data and the ability to discern more specific patterns.

#### 2.3.2. Knowledge Transfer from Adult to Pediatric CXR

To better learn the domain knowledge and adapt the previously learned adult CXR distribution to pediatric CXRs, a further transfer-learning phase specifically for pediatric CXR images was performed. This final stage involves training the already pre-trained and fine-tuned ViT encoder using the PediCXR dataset’s training set, followed by its evaluation using the test set. We evaluated the proposed approach under two scenarios: (a) a linear classification setting where the pre-trained encoder weights were frozen but the linear classification head on top was trainable for the task and (b) a fine-tuning setting where both the encoder and the linear head were fine-tuned. For the (a) setting, following and adapted from the protocol in [12], a LARS optimizer was applied with momentum = 0.9 and WeightDecay = 0.05. The encoder was trained with 100 epochs at LearningRate = 0.1 and BatchSize = 128. For the (b) setting, we utilized the AdamW optimizer ($\beta_{1}$ = 0.9, $\beta_{2}$ = 0.95, WeightDecay = 0.05). The LearningRate was established at 2.5e-4 and a BatchSize of 128. This process extended across a total of 75 epochs. All experiments were run three times with different random seeds.

#### 2.3.3. Model Evaluation

For model comparison, our primary focus was on the mean area under the receiver operating characteristic curve (AUC) across all classes (mAUC), as well as the AUC for each of the six individual classes. In Section 3.6, we utilized specificity, sensitivity, and F1 score to evaluate the models’ discriminative ability for positive and negative classes of each disease label. The metrics are defined as follows:$$\begin{array}{cc} \;\mathrm{Precision}\; & =\frac{TP}{\left(TP+FP\right)}\\ \;\mathrm{sensitivity}\; & =\frac{TP}{TP+FN}\\ \;\mathrm{Specificity}\; & =\frac{TN}{TN+FP}\\ \;F1\;\mathrm{Score}\; & =\frac{2\times \;\mathrm{sensitivity}\;\times \;\mathrm{Precision}\;}{\;\mathrm{sensitivity}\;+\;\mathrm{Precision}\;} \end{array}$$

The effectiveness of clustering image embeddings was evaluated using the Davies–Bouldin Index (DBI). Lower DBI values suggest that similar images are more effectively clustered. The formula is:$$DBI=\frac{1}{k}\sum_{i=1}^{k}\max_{j\neq i}\left(\frac{s_{i}+s_{j}}{d_{ij}}\right)$$
where *k* is the number of clusters, $s_{i}$ is the average distance of all points in cluster *i* to the centroid of cluster *i* (intra-cluster distance), and $d_{ij}$ is the distance between the centroids of clusters *i* and *j* (inter-cluster distance).

## 3. Results

### 3.1. Model Performance with Supervised Learning

We first started with the baseline models and directly trained the pediatric CXR using supervised learning with random initialization. The results are summarised in Table 3. The table shows that DenseNet121 performed the best, outperforming ResNet 50, ViT-Small/16, and ViT-Base/16. DenseNet121 achieved an average mAUC of 0.714 (95% CI: 0.709–0.719), and the AUC for broncho-pneumonia, bronchiolitis, bronchitis, no finding, pneumonia, and other diseases are 0.781 (95% CI: 0.780–0.782),0.710 (95% CI: 0.703–0.717), 0.698 (95% CI: 0.696–0.700), 0.726 (95% CI: 0.721–0.731), 0.744 (95% CI: 0.732–0.756), and 0.625 (95% CI:0.613–0.637), respectively. ViTs do not perform as well as CNNs. This is possibly because ViTs generally require much larger datasets to achieve optimal performance compared to CNNs. CNNs, like DenseNet, have inherent inductive biases such as translation invariance and local connectivity, allowing for better generalizability on small datasets like PediCXR. Additionally, we also report the previously published results using the same dataset [41] for reference.

### 3.2. Transfer Learning via Linear Evaluation

The results of the linear evaluation for both ViT-Small/16 and ViT-Base/16 are summarized in Table 4. The three-stage approach increased the model’s performance remarkably compared to its supervised training counterpart. In the case of ViT-Small/16, the most effective model was the one fine-tuned using the CheXpert dataset, which resulted in a 2.5% improvement in mAUC. Similarly, for ViT-Base/16, fine-tuning with the CheXpert dataset yielded the best model, achieving a 5.5% increase in the mAUC. Another observation is that the larger models (ViT-Base/16) always perform better than the smaller ones(ViT-Small/16).

### 3.3. Transfer Learning via Fine-Tuning

In contrast with linear evaluation, the fine-tuning setting more faithfully mirrors the real-world application of pre-trained encoders. In this context, we provide a more comprehensive analysis of fine-tuning evaluation results, which are detailed in Table 5.

Compared to the results listed in the last two sections, the major finding about ViT models is that fine-tuned models perform better than the corresponding linearly evaluated and supervised trained ones. Investigating the data in the table alone, it is evident that the ViT-Base/16 encoders exhibit superior performance in comparison to the ViT-Small/16 encoders, though the smaller ones also deliver acceptable results. The ViT-Base/16 fine-tuned with CheXpert shows the best model performance on the PediCXR dataset with an mAUC of 0.761. This performance surpasses models that were only pre-trained with X-ray data or fine-tuned with COVIDx and ChestX-ray14 data by 2.4%, 1.7% and 0.4%, respectively. Additionally, this particular model yields the highest performance in diagnosing bronchitis (0.7413), pneumonia (0.8350), and other rare diseases (0.6833). The ViT-Small/16 encoder pre-trained on the ImageNet dataset performed the worst with an mAUC of only 0.719 and that of the encoder pre-trained on the CXR data improved by a small margin. The performance of the ViT-Small/16 encoder improved significantly when it underwent a two-stage training and additional fine-tuning on the pediatric CXR dataset (2.6%, 2.3%, and 2.6% when fine-tuned with CheXpert, COVIDx, and ChestX-ray14, respectively). Interestingly, fine-tuning with the COVIDx dataset yielded minimal improvements in model performance. This may be attributed to the distinct disease distribution in COVIDx compared to that in PediCXR, which constrains the model’s capacity to generalize across different datasets effectively.

One key insight about ViT models emerges from this study. ViT models pre-trained on CXRs, then fine-tuned on adult CXR datasets, and further fine-tuned on pediatric CXR data outperform those only pre-trained on general X-rays and directly fine-tuned on pediatric CXR data. This enhanced performance can be attributed to the broad knowledge acquired from supervised learning using adult CXR, which is beneficial in adapting to the nuances of pediatric CXR data, especially given the limited size of the PediCXR dataset.

DenseNet121’s performance on PediCXR holds a different story. The model achieves a higher mAUC of 0.749 without fine-tuning on adult CXR data, compared to a lower mAUC of 0.713 when fine-tuning is applied. This indicates that fine-tuning DenseNet121 on adult CXR might lead to overfitting to adult-specific features, consequently diminishing its performance on pediatric CXR images.

### 3.4. Model Interpretation

Grad-CAM was utilized to visualize the CXRs, specifically to highlight the areas indicative of disease through bright colors. Figure 2 shows saliency maps of the sampled images from the PediCXR test set for all models presented in Section 3.3. Observations from Figure 2 reveal that models pre-trained on the ImageNet dataset fail to localize the pertinent regions accurately, whereas those pre-trained on CXR datasets achieve slightly better localization. In comparison, models pre-trained on CXRs, with an infusion of domain-specific knowledge, can pinpoint potential areas of interest. Nonetheless, models based on the Vit-Small/16 architecture fail to delineate the affected regions precisely. For instance, as depicted in Figure 2c, the model fine-tuned with CheXpert data erroneously highlights the spinal area, whereas the one fine-tuned with COVIDx data emphasizes both the left and right lung areas, deviating from the actual ground truth. Conversely, models employing the Vit-Base/16 architecture show enhanced performance, with the region identified by the model fine-tuned using CheXpert data aligning closely with the ground truth.

### 3.5. Embedding Visualization

To better understand how various initialization methods differentiate between disease categories in the embedding space, we present *t*-distributed stochastic neighbor (t-SNE) visualizations of PediCX training samples for four representative ViT encoders in Figure 3 along with the Davies–Bouldin index noted in the subplot title. Notably, compared to natural image classification, this task is more complex due to the high inter-class similarity in medical imaging data and the inherently noisy nature of the labels. As the figure suggests, the encoder fine-tuned with CheXpert data demonstrates marginally superior performance to the others, as evidenced by the more cohesive clustering of data from the same class and the smallest Davies–Bouldin index.

### 3.6. Error Analysis with the Best-Performance Model

Table 6 presents the comprehensive evaluation metrics for each disease category using the top-performing ViT model. With an optimized threshold setting, the model attains an average sensitivity of 0.639, a specificity of 0.683, and an F1 score of 0.376 in the test dataset. In the case of ’Bronchitis,’ the model successfully identifies most young patients affected (72.4%), showing similar sensitivity for broncho-pneumonia and pneumonia. However, its performance is worse for bronchiolitis and other diseases. This could be attributed to the diagnostic challenges associated with bronchiolitis, as discussed in [42], and the significant variability within the ’Other disease’ category, which complicates the model’s ability to discern specific patterns necessary for accurate decision-making. Additionally, our model shows quite high specificity scores across all disease categories, suggesting that the model can correctly recognize disease-free patients.

## 4. Discussion

Here, we present a three-stage transfer learning system designed to accurately diagnose pediatric chest X-rays (CXRs) and classify them into subtypes. Our approach involves leveraging models pre-trained and fine-tuned on adult CXRs, followed by further fine-tuning on pediatric CXRs to enhance overall performance. The model’s performance on the testing dataset demonstrates several key points:Introducing prior knowledge through masked autoencoder (MAE) pre-training on adult CXRs significantly boosts model performance on pediatric CXRs.Fine-tuning on adult CXRs further enhances the model’s ability to learn the intricate characteristics of thoracic disease distributions.Larger vision transformer models fine-tuned on the CheXpert dataset exhibit the best performance, both quantitatively and qualitatively.

The proposed three-stage system—consisting of masked autoencoder pre-training on adult X-ray images, fine-tuning on the CheXpert dataset, and final fine-tuning on a pediatric dataset—demonstrates a superior performance by progressively refining the model’s feature representations. This approach outperforms directly supervised learning and a two-stage fine-tuning system by leveraging robust feature learning, effective domain adaptation, and targeted specialization. The initial pre-training stage establishes a strong foundation, the CheXpert fine-tuning enhances the model’s understanding of thoracic diseases, and the final stage ensures the model is finely tuned to the unique characteristics of pediatric X-rays, resulting in a more accurate and generalizable model. Pediatric chest radiographs capture the same body segment with the same anatomical structures as adult chest radiographs, so pre-training with larger adult datasets is a promising approach when developing models with limited pediatric training data [43,44]. However, the differing body size, the relative size of the structures (the cardiac silhouette is proportionally larger relative to the thorax in children for example), and the differing disease processes that affect children and adults mean that training and validation on pediatric images is of the same critical.

A three-stage system using ViT-Base/16 as the backbone demonstrates the best performance on the PediCXR dataset. ViT-Base generally outperforms ViT-Small, likely due to its larger number of parameters, which enable it to capture more intricate patterns and nuances in the data. The fine-tuned CheXpert dataset performs better than other datasets, probably because it encompasses a wider variety of thoracic disease conditions. This variety allows models to learn more complex features and patterns compared to the more homogeneous COVIDx dataset.

Additionally, our error analysis identifies specific diseases where the system excels, achieving high sensitivity and F1 scores. Notably, the model shows a strong performance in identifying pneumonia, one of the most clinically relevant respiratory diseases in children [45]. Consequently, our model demonstrates a robust overall performance, particularly in diagnosing pneumonia, and holds potential for further improvement in less severe diseases.

This work has several limitations. The current diagnosis may not fully reflect its clinical significance. The current dataset was collected from two major hospitals in Vietnam and annotated by their seventeen radiologists [25]. Therefore, the clinical definition might vary slightly among different annotators and may also differ from the data on which it could be deployed. Bronchiolitis is a clinical diagnosis and cannot be determined solely based on a CXR [46]. Furthermore, bronchitis is generally not a diagnosis in children [47]. Therefore, the clinical relevance and actionability of these labels are likely limited. Additionally, the annotations in the PediCXR dataset are subjective, with discrepancies among different annotators. Increasing the number of annotators could enhance the quality of CXR labeling and validate model performance. To improve the model’s generalizability and practicality, efforts include (1) establishing clinical labels with high clinical relevance, (2) collecting more annotated data, (3) utilizing multiple annotators and incorporating label uncertainty into the network to reduce labeling bias and improve model robustness, and (4) validating model performance on external datasets can be made.

In summary, the main contributions of our work include the following:We developed a three-stage training system and assessed its effectiveness in classifying thoracic diseases using the most recent and extensive publicly accessible pediatric CXR dataset, PediCXR.Our study involved extensive quantitative experiments including (a) comparisons with direct training with a supervised learning strategy; (b) the use of CXR or non-medical images for pre-training; (c) the utilization of various adult CXR datasets for fine-tuning; and (d) an evaluation with linear and fine-tuning settings. Our findings demonstrate quantitatively and qualitatively that the top-performing model is the ViT-Base/16. This model was pre-trained on CXR, fine-tuned on CheXpert, and then further fully fine-tuned on PediCXR.We performed a detailed error analysis on PediCXR using the best-performing model, thoroughly examining its performance in identifying common diseases.
Although the proposed methods are preliminary and not ready for immediate deployment, these early results support the potential for using these three-stage decision support systems for pediatric thoracic disease diagnosis. Compared to manually labeling the CXRs and gathering a board of certified radiologists, this system can provide a more cost-effective method for use in therapeutic trials, research, and clinical practice. The key to this method lies in (1) using millions of public adult CXRs for pre-training to improve the model’s capability of learning the CXRs and their relevant disease distribution and (2) shifting the distribution learned on adults toward children. In addition, other clinical features, including the electronic health records data that are routinely collected, should also be considered to gain a deeperr understanding of the disease’s progression and mechanism, therefore allowing for better diagnosis and treatment [48]. There are works utilizing multi-modal learning for adult thoracic disease early detection or diagnosis [49,50], and these works can also be extended to children.

## 5. Conclusions

This study introduces a three-stage automated clinical support system for precise diagnostics in pediatric CXRs. It employs a masked autoencoder pre-trained on the large-scale adult CXR datasets, followed by knowledge transfer to adapt to pediatric cases. Based on the various experiments, it can be seen that models pre-trained on CXR and subsequently fine-tuned significantly outperform other models in overall performance and diseased region localization. Notably, models pre-trained on CXRs and fine-tuned using the CheXpert dataset demonstrated the best performance, with disease localization closely aligning with the ground truth. In the future, we will further explore other pre-training schemes, including BYOL [10], DINO [11], and use bounding box coordinates to improve the model performance and enhance disease region localization on external datasets.

## Figures and Tables

**Figure 2 diagnostics-14-01634-f002:**
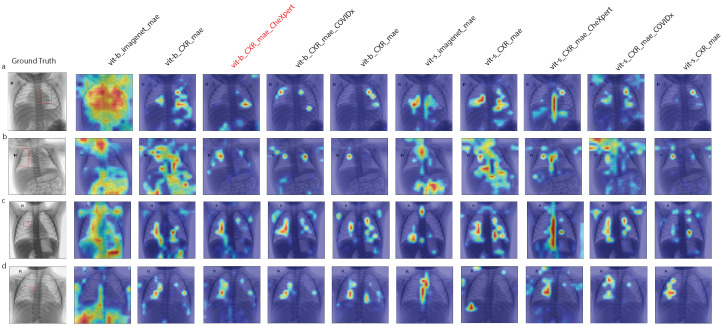
**Grad-CAM visualizations on four pediatric CXR samples**. The first column on the left, featuring (**a**–**d**) as four randomly drawn diseased samples, displays the original CXR, recognized as the ground truth, with the diseased areas highlighted in red boxes. The subsequent columns showcase saliency maps created with various initializations overlaying on the original X-ray images. The bright colors signify areas of relevance to the model’s predictions.

**Figure 3 diagnostics-14-01634-f003:**
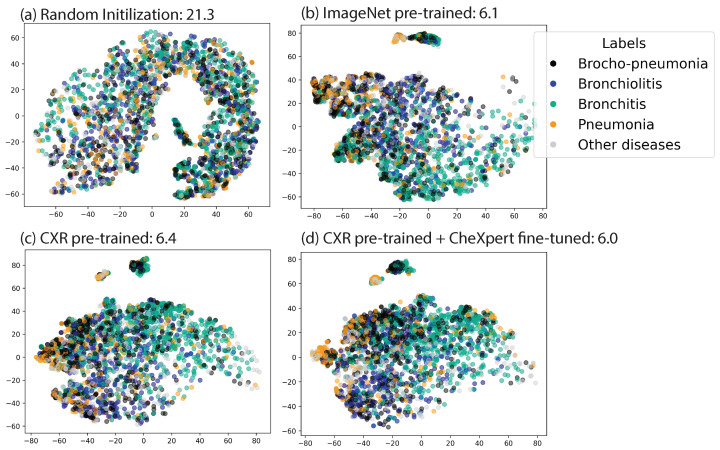
**t-SNE comparison of image representations from ViT-Base/16 models (DBI is presented along with the title)**: (**a**) supervised training with random initialization; (**b**) pre-trained on ImageNet using MAE; (**c**) pre-trained on adult CXR using; (**d**) pre-trained on adult CXR with MAE and subsequently fine-tuned using CheXpert data.

**Table 1 diagnostics-14-01634-t001:** Brief summary of different approaches for image pre-training.

Strategies	Methods (Ref.)
Innate relationship	Image rotation prediction [16]; Image context prediction [17]
Generative	Autoencoder [18]; GAN [19]
Contrastive	SimCLR [20]; BYOL [10]; DINO [11]
Self-prediction	MAE [12]

**Table 2 diagnostics-14-01634-t002:** Numbers of chest X-rays in the training and test Sets of PediCXR.

Dataset	Samples (n)	Brocho-Pneumonia (n)	Bronchiolitis (n)	Bronchitis (n)	Pneumonia (n)	No Finding (n)	Other Diseases (n)
Training	7728	545	497	842	392	5143	463
Test	1387	84	90	174	89	907	81

**Table 3 diagnostics-14-01634-t003:** Supervised training on PediCXR with random initialization: mean (std). The highest mAUC scores are bold. The *p*-value indicates the statistical significance of the mAUC achieved by direct supervised learning methods compared to the best-performing model using the proposed three-stage system. †: The results with the same dataset are reported in [41].

Encoder	mAUC	AUC for Every Class						
		Broncho-Pneumonia	Bronchiolitis	Bronchitis	No Finding	Pneumonia	Others	*p*-Value
DenseNet121 †	0.709	0.696	0.638	0.691	0.776	0.802	0.703	<0.0001
Densenet121	**0.714**	**0.781**	**0.710**	**0.698**	**0.726**	**0.744**	**0.625**	<0.0001
	**[0.709–0.719]**	**[0.780–0.782]**	**[0.703–0.717]**	**[0.696–0.700]**	**[0.721–0.731]**	**[0.732–0.756]**	**[0.613–0.637]**	
ResNet50	0.685	0.736	0.690	0.686	0.696	0.709	0.592	<0.0001
	[0.679–0.691]	[0.727–0.745]	[0.676–0.704]	[0.685–0.687]	[0.691–0.701]	[0.693–0.725]	[0.577–0.607]	
ViT–Small/16	0.646	0.698	0.627	0.646	0.655	0.652	0.600	<0.0001
	[0.641–0.652]	[0.696–0.699]	[0.621–0.632]	[0.644–0.649]	[0.648–0.662]	[0.643–0.661]	[0.580–0.621]	
ViT–Base/16	0.654	0.704	0.649	0.648	0.657	0.655	0.609	<0.0001
	[0.648–0.659]	[0.694–0.715]	[0.632–0.665]	[0.643–0.654]	[0.655–0.658]	[0.636–0.674]	[0.607–0.612]	

**Table 4 diagnostics-14-01634-t004:** Linear evaluation on PediCXR: mean (std). The highest mAUC scores are bold. The *p*-value indicates the statistical significance of the mAUC achieved by the methods compared to the best-performing model using the proposed three-stage system.

Encoder	Pretrained	Finetuned	mAUC	AUC for Every Class						
				Broncho-Pneumonia	Bronchiolitis	Bronchitis	No Finding	Pneumonia	Others	*p*-Value
ViT-Small/16	X-ray	COVIDx	0.643	0.670	0.611	**0.679**	0.695	0.667	0.538	<0.0001
			[0.631–0.656]	[0.645–0.695]	[0.557–0.665]	**[0.676–0.682]**	[0.690–0.701]	[0.605–0.729]	[0.512–0.565]	
		X-ray	0.653	0.700	0.637	0.647	0.696	0.695	0.545	<0.0001
			([0.648–0.658]	[0.689–0.710]	[0.615–0.659]	[0.641–0.653]	[0.689–0.703]	[0.664–0.727]	[0.518–0.572]	
		CheXpert	0.662	0.707	0.632	0.655	0.690	0.734	0.554	<0.0001
			[0.662–0.662]	[0.675–0.740]	[0.625–0.640]	[0.647–0.663]	[0.683–0.697]	[0.723–0.744]	[0.518–0.590]	
ViT-Base/16	X-ray	COVIDx	0.675	0.710	0.657	0.674	0.704	0.711	0.597	<0.0001
			[0.662–0.689]	[0.682–0.739]	[0.616–0.698]	[0.649–0.698]	[0.699–0.708]	[0.674–0.747]	[0.551–0.643]	
		X-ray	0.678	0.739	0.658	0.646	0.702	0.735	0.586	<0.0001
			[0.673–0.682]	[0.723–0.755]	[0.648–0.669]	[0.628–0.664]	[0.698–0.705]	[0.727–0.744]	[0.553–0.620]	
		CheXpert	**0.690**	**0.745**	**0.677**	0.670	**0.708**	**0.737**	**0.604**	<0.0001
			**[0.682–0.698]**	**[0.728–0.762]**	**[0.642–0.712]**	[0.654–0.686]	**[0.703–0.712]**	**[0.722–0.752]**	**[0.578–0.631]**	

**Table 5 diagnostics-14-01634-t005:** Fine-tuning evaluation on PediCXR: mean(std). The highest mAUC scores are bold. The *p*-value indicates the statistical significance of the mAUC achieved by the best performing model, ViT-Base/16 fine-tuned on CheXpert, in comparison to other methods.

Encoder	Pretrained	Finetuned	mAUC	AUC for Every Class						
				Broncho-Pneumonia	Bronchiolitis	Bronchitis	No Finding	Pneumonia	Others	*p*-Value
DenseNet121	X–ray	–	0.749	0.823	0.732	0.722	**0.776**	0.827	0.615	<0.0001
			[0.747–0.752]	[0.817–0.830]	[0.729–0.734]	[0.720–0.724]	**[0.770–0.783]**	[0.818–0.836]	[0.610–0.620]	
		X–ray	0.713	0.779	0.709	0.700	0.725	0.744	0.621	<0.0001
			[0.708–0.717]	[0.772–0.786]	[0.701–0.717]	[0.695–0.704]	[0.724–0.725]	[0.735–0.754]	[0.602–0.639]	
ViT–Small/16	Imagenet	–	0.719	0.787	0.709	0.711	0.729	0.761	0.618	<0.0001
			[0.716–0.723]	[0.780–0.794]	[0.709–0.710]	[0.707–0.715]	[0.725–0.732]	[0.758–0.765]	[0.613–0.624]	
		–	0.729	0.808	0.721	0.708	0.744	0.770	0.626	<0.0001
			[0.727–0.732]	[0.806–0.810]	[0.717–0.725]	[0.707–0.709]	[0.740–0.748]	[0.764–0.776]	[0.617–0.636]	
	X–ray	COVIDx	0.746	0.825	0.725	0.729	0.760	0.797	0.639	<0.005
			[0.741–0.751]	[0.818–0.833]	[0.716–0.733]	[0.726–0.732]	[0.756–0.764]	[0.779–0.815]	[0.632–0.645]	
		X–ray	0.748	0.824	0.725	0.737	0.761	0.798	0.642	<0.0001
			[0.747–0.748]	[0.820–0.827]	[0.719–0.731]	[0.732–0.741]	[0.760–0.762]	[0.797–0.800]	[0.640–0.643]	
		CheXpert	0.748	0.818	0.719	0.740	0.758	0.805	0.650	<0.0001
			[0.748–0.749]	[0.817–0.819]	[0.718–0.720]	[0.737–0.742]	[0.757–0.759]	[0.800–0.810]	[0.645–0.654]	
ViT–Base/16	Imagenet	–	0.746	0.824	0.729	0.722	0.759	0.809	0.633	<0.005
			[0.745–0.747]	[0.823–0.826]	[0.727–0.732]	[0.720–0.725]	[0.756–0.761]	[0.807–0.810]	[0.631–0.634]	
	X–ray	–	0.743	0.818	0.728	0.722	0.757	0.800	0.633	0.2
			[0.740–0.746]	[0.812–0.824]	[0.723–0.733]	[0.721–0.724]	[0.754–0.759]	[0.792–0.808]	[0.628–0.639]	
		COVIDx	0.750	**0.833**	0.732	0.725	0.762	0.814	0.634	<0.005
			[0.749–0.751]	**[0.830–0.835]**	[0.730–0.734]	[0.720–0.730]	[0.760–0.765]	[0.812–0.816]	[0.628–0.641]	
		X–ray	0.760	0.825	**0.733**	0.726	0.767	0.831	0.678	<0.0001
			[0.758–0.761]	[0.824–0.827]	**[0.729–0.736]**	[0.721–0.730]	[0.765–0.769]	[0.827–0.835]	[0.673–0.683]	
		CheXpert	**0.761**	0.831	0.711	**0.741**	0.766	**0.835**	**0.683**	
			**[0.760–0.763]**	[0.829–0.833]	[0.709–0.714]	**[0.737–0.745]**	[0.764–0.767]	**[0.833–0.837]**	**[0.672–0.695]**	

**Table 6 diagnostics-14-01634-t006:** Detailed model performance on PediCXR test dataset with ViT-Base/16 model fine-tuned on CheXpert.

Label	Accuracy	Sensitivity	Precision	Specificity	F1 Score
Brocho-pneumonia	0.801	0.691	0.187	0.801	0.294
Bronchiolitis	0.765	0.500	0.137	0.784	0.215
Bronchitis	0.632	0.724	0.213	0.619	0.329
Pneumonia	0.894	0.618	0.325	0.913	0.426
Other diseases	0.852	0.309	0.142	0.885	0.195
mean	0.770	0.639	0.279	0.683	0.376

## Data Availability

The source code is available at https://github.com/kayvanlabs/PediCXR-MAE (accessed on 21 June 2024) and the data can be obtained from https://physionet.org/content/vindr-cxr/1.0.0/ (accessed on 21 June 2024).

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
