# Peer review of "Three-Stage Frameworkfor Accurate Pediatric Chest X-ray Diagnosis Using Self-Supervision and Transfer Learning on Small Datasets"

_diagnostics, 2024, doi:10.3390/diagnostics14151634_

Round 1

Reviewer 1 Report

Comments and Suggestions for Authors

The interpretation of the chest radiograph can be challenging due to the superimposition of anatomical structures along the projection direction. This effect can make it very difficult to detect abnormalities in particular locations, to detect small or subtle abnormalities, or to accurately distinguish between different pathological patterns. The authors proposed a three-stage framework with knowledge transfer from adult chest X-rays to aid the diagnosis and interpretation of pediatric thorax diseases.

Comments

1.      The abstract should provide the results of the proposed model, along with a quantitative comparison to state-of-the-art approaches.

2.      Including the summary of the literature review in tabular form could enhance clarity and organization.

3.      In addition to Precision, Recall, Specificity, and F1 Score, other parameters may be used to evaluate the proposed system.

Comments on the Quality of English Language

Minor editing of English language required

Author Response

Reviewer 1

The interpretation of the chest radiograph can be challenging due to the superimposition of anatomical structures along the projection direction. This effect can make it very difficult to detect abnormalities in particular locations, to detect small or subtle abnormalities, or to accurately distinguish between different pathological patterns. The authors proposed a three-stage framework with knowledge transfer from adult chest X-rays to aid the diagnosis and interpretation of pediatric thorax diseases.

Comments

  1.     The abstract should provide the results of the proposed model, along with a quantitative comparison to state-of-the-art approaches.

Response: Thank you for your suggestion. We added the comparison in the abstract. ‘The baseline models, ViT-Small/16 and ViT-Base/16, when directly trained on the Pediatric CXR dataset, only achieved mean AUC scores of 0.646 (95\% CI: 0.641 – 0.651) and 0.654 (95\% CI: 0.648 – 0.660), respectively.’ 

  1.     Including the summary of the literature review in tabular form could enhance clarity and organization.

Response: Thank you for your suggestion. We include a brief literature review in tabular format in the introduction to illustrate several self-supervised strategies that have been used in the areas of CXR and medical imaging.

  1.   In addition to Precision, Recall, Specificity, and F1 Score, other parameters may be used to evaluate the proposed system.

Response: Thank you for your advice. Apart from precision, recall, and specificity, we provided additional metrics, including accuracy and precision, in the table to further evaluate the proposed system.

Emily Wittrup and Dr. Joseph Kohne, being native speakers, further refined and edited the manuscript.

Reviewer 2 Report

Comments and Suggestions for Authors

The aim of this work was to develop a powerful model for evaluating the accuracy of pediatric radiographic diagnosis.

I have some suggestions.

TITLE: Please report on the study design

ABSTRACT:

Report 95%CI along with the AUC values.

Use the term “sensitivity” and not “recall” (use the term “sensitivity” throughout the manuscript).

INTRO:

Lines 65-77: Please move this part to the end of the manuscript before the conclusions section after the limitations.

Please state the aim of the study.

RESULTS: Please provide the 95%CI for each AUC in both the tables and the main text

I suggest adding AUC comparison tests (difference between AUC).

DISCUSSION: I think the explanation of the results could be deepened in the light of current knowledge (e.g. why the ViT-Base/16 model performs better than others, why this model can be better fitted to pediatric data than others, etc.). The results should be discussed.

Also explain why the methodological approach chosen by the authors (pre-training with adult CXR data) provides good evidence for the development of pediatric CXR data (e.g., explain the similarities and differences between the two types of data).

Lines 209-302: please move to the "Conclusion" section.

Author Response

The aim of this work was to develop a powerful model for evaluating the accuracy of pediatric radiographic diagnosis.

I have some suggestions.

Comments 1:

TITLE: Please report on the study design

Response 1: Thank you for your suggestions. We change from ‘Self-Supervision and Transfer Learning for an Accurate Pediatric Chest X-Ray Diagnosis with Small Dataset’ to ‘Three-Stage Framework for Accurate Pediatric Chest X-Ray Diagnosis Using Self-Supervision and Transfer Learning on Small Datasets’

Comments 2:

ABSTRACT:

Report 95%CI along with the AUC values.

Use the term “sensitivity” and not “recall” (use the term “sensitivity” throughout the manuscript).

Response 2: Thank you for pointing it out. We reported 95% CI along with AUC in the abstract for easy of comparison. The corresponding sentences are revised as ‘The ViT-Base/16 model, fine-tuned with the CheXpert dataset, a large chest X-ray dataset, emerged as the most effective, achieving a mean AUC of 0.761 (95\% CI: 0.759 – 0.763) across six disease categories and demonstrating high sensitivity (average 0.639) and specificity (average 0.683), indicative of its strong discriminative ability. The baseline models, ViT-Small/16 and ViT-Base/16, when directly trained on the Pediatric CXR dataset, only achieved mean AUC scores of 0.646 (95\% CI: 0.641 – 0.651) and 0.654 (95\% CI: 0.648 – 0.660), respectively.’In addition, to ensure consistency in the term, we used ‘sensitivity’ throughout the manuscript.

Comments 3:

INTRO:

Lines 65-77: Please move this part to the end of the manuscript before the conclusions section after the limitations.

Response 3: Thank you for the feedback. We moved this to the end of the manuscript before the conclusions section after the limitations and integrated it with the previous last paragraph in the discussion section.

Comments 4:

Please state the aim of the study.

Response 4:

We added this sentence and modified the corresponding contexts in the introduction section. ‘In this study, we aim to create an accurate automated system for diagnosing pediatric chest X-rays (CXR), addressing the challenge of limited pediatric CXR data. To achieve this, we develop a three-stage transfer learning approach specifically for classifying pediatric CXR diagnoses.’

Comments 5:

RESULTS: Please provide the 95%CI for each AUC in both the tables and the main text

I suggest adding AUC comparison tests (difference between AUC).

Response: Thank you for your suggestion. We included the 95% confidence scores in every result table. Additionally, in results tables, we included an AUC comparison test, offering a statistical significance analysis to demonstrate that the proposed method is statistically superior.

Comments 5:

DISCUSSION: I think the explanation of the results could be deepened in the light of current knowledge (e.g. why the ViT-Base/16 model performs better than others, why this model can be better fitted to pediatric data than others, etc.). The results should be discussed.

Response: Thank you for your suggestions. We added a paragraph in the discussion section to further discuss why ViT-Base/16 model is better: ‘A three-stage system using ViT-Base/16 as the backbone demonstrates the best performance on the PediCXR dataset. ViT-Base generally outperforms ViT-Small, likely due to its larger number of parameters, which enable it to capture more intricate patterns and nuances in the data. The fine-tuned CheXpert dataset performs better than other datasets, probably because it encompasses a wider variety of thoracic disease conditions. This variety allows models to learn more complex features and patterns compared to the more homogeneous COVIDx dataset.’

Comments 6:

Also explain why the methodological approach chosen by the authors (pre-training with adult CXR data) provides good evidence for the development of pediatric CXR data (e.g., explain the similarities and differences between the two types of data).

Response 6: Thank you for your suggestions. We added one paragraph in the discussion section discussion why the three-stage system is better than direct supervised learning and other methods. ‘The proposed three-stage system—consisting of masked autoencoder pretraining on adult X-ray images, fine-tuning on the CheXpert dataset, and final fine-tuning on a pediatric dataset—demonstrates superior performance by progressively refining the model's feature representations. This approach outperforms direct supervised learning and a two-stage fine-tuning system by leveraging robust feature learning, effective domain adaptation, and targeted specialization. The initial pretraining stage establishes a strong foundation, the CheXpert fine-tuning enhances the model's understanding of thoracic diseases, and the final stage ensures the model is finely tuned to the unique characteristics of pediatric X-rays, resulting in a more accurate and generalizable model.’

Comments 7:

Lines 209-302: please move to the "Conclusion" section.

Response 7:

Thank you for your feedback. To address your concern, I have moved the paragraph to clarify its purpose.

Round 2

Reviewer 2 Report

Comments and Suggestions for Authors

Good job. The manuscript is now suitable for publication